# Physicochemical Properties and Cytoprotective Effects on PC12 Cells of Polysaccharides from *Belamcanda chinensis* (L.) DC. Obtained via a Gradient Ethanol Precipitation Method

**DOI:** 10.3390/molecules30050998

**Published:** 2025-02-21

**Authors:** Yuanqi Duan, Jinfeng Sun, Yongkang Xue, Weiwei Xu, Yuxin Jiang, Tieqiang Zong, Wei Zhou, Zhengyu Hu, Gao Li

**Affiliations:** Key Laboratory of Natural Medicines of the Changbai Mountain, Ministry of Education, College of Pharmacy, Yanbian University, Yanji 133002, China; duanyuanqi@163.com (Y.D.); jfsun@ybu.edu.cn (J.S.); xueykang@163.com (Y.X.); xuweiwei6780121@163.com (W.X.); tina841623394@163.com (Y.J.); ztq405955580@163.com (T.Z.); zw2015@ybu.edu.cn (W.Z.)

**Keywords:** *Belamcanda chinensis* (L.) DC rhizomes, gradient ethanol precipitation method, polysaccharides, ischemic stroke

## Abstract

Given that the preparation method of polysaccharides affects the functional properties, four types of acidic polysaccharides (BCP30-1a, BCP50-1a, BCP70-1a, and BCP90-1a) were prepared using the gradient ethanol precipitation method. Then, a series of chemical and instrumental analysis techniques were used to compare structural characteristics and morphology. Neuroprotective effects were explored using OGD/R-induced PC12 cells. The results showed that BCP30-1a, BCP50-1a, BCP70-1a, and BCP90-1a had similar characteristic groups and contained both β-glycosidic and α-glycosidic bonds. Their molecular weights, in descending order, were 198.398 kDa, 184.690 kDa, 184.556 kDa, and 184.217 kDa, respectively. In addition, the four polysaccharides contained different proportions of glycosidic bonds, namely, Man*p*-(1→, →5)-Ara*f*-(1→, →3)-Gal*p* (or GalA*p*)-(1→, →4)-Glc*p*-(1→ and →3,6)-Gal*p*-(1→. BCP30-1a also contained a certain proportion of Gal*p*-(1→, and each polysaccharide had different microscopic characteristics and good thermal stability. Finally, BCP50-1a, BCP70-1a, and BCP90-1a exhibited good cytoprotective effects on PC12 cells based on the OGD/R model. These findings provide a novel regulatory strategy for the functional activity of BCPs and offer scientific evidence supporting application in the research field of ischemic stroke.

## 1. Introduction

Ischemic stroke (IS), stemming from a sudden cessation of blood flow to the brain, constitutes a grave neurological disorder leading to neuronal damage and functional impairment [1]. Globally, stroke continues to rank as the second leading cause of mortality and the third significant contributor to long-term disability [2]. When an ischemic stroke occurs, the brain tissue experiences acute oxygen and nutrient deprivation as a result of the abrupt cessation of blood supply, causing cellular injury and eventual necrosis [3]. The primary pathogenesis of stroke involves ischemia–reperfusion injury. Although reperfusion therapy can temporarily restore blood flow during the acute phase, it may also induce secondary injury mechanisms that exacerbate ischemic injury [4]. This ischemia–reperfusion process induces significant oxidative injury to surrounding neurons, which can be mimicked in vitro by an oxygen and glucose deprivation (OGD)/re-oxygenation (OGD/R) procedure applied to cultured neurons [5]. The OGD/R model is commonly used to investigate the effects of ischemia–reperfusion injury at the cellular level. In vitro cell culture experiments show that OGD/R can induce two types of cell death processes, namely, pyroptosis and apoptosis, which have similar outcomes but are entirely distinct in their underlying mechanisms [6]. PC12 cells treated with OGD/R have been widely used to simulate ischemia–reperfusion injury in vitro. It is an ideal cell model for studying IS that can simulate the process of cell injury in vivo after ischemia and hypoxia by depriving cells of nutrients and oxygen in a short period of time, followed by reoxygenation with glucose [7,8]. It offers the advantages of being closer to the actual state of ischemia–reperfusion injury, having fewer interfering factors, exhibiting reliable effects, and demonstrating strong reproducibility. Consequently, there is an urgent need to investigate innovative and alternative cytoprotective methods and treatment strategies based on the OGD/R model.

In traditional Chinese medicine theory, the pathogenesis of IS is often related to factors such as pyretic toxicity, blood stagnation, and turbid phlegm. Therefore, clearing heat and detoxification, promoting blood circulation to remove blood stasis, and dissipating phlegm are the main treatment principles for acute stroke. For example, Ma et al. [9], Tu et al. [10], and Yang et al. [11] studied the relationship between traditional Chinese medicines such as *Scrophularia ningpoensis*, *Ginkgo biloba* leaf, and *Salvia miltiorrhiza* and IS. The results showed that these herbs indeed have good preventive and therapeutic effects against IS. This indicated that traditional Chinese medicine with these effects has broad development prospects in the field of IS. The traditional Chinese medicine, *Belamcanda chinensis* (L.) DC., with its traditional effects of clearing heat, removing blood stasis, and resolving phlegm [12], has shown promising prospects as a potential therapeutic drug for IS. *B. chinensis* is the sole species in the genus *Belamcanda* Adans. Contemporary pharmacological research has demonstrated that *B. chinensis* exhibits diverse biological properties, including anti-inflammatory, analgesic, antibacterial, antiviral, antioxidant, anticancer, and estrogenic activities, thereby possessing significant potential for a broad spectrum of clinical applications [13,14,15]. Polysaccharides isolated from *B. chinensis* (BCPs), which are the main active ingredients in traditional Chinese medicine, have been confirmed by Duan et al. [16] and Zhao et al. [17] to possess good anti-complementary and anti-tumor pharmacological activities. However, there are few reports on their application in IS, even though traditional use suggests that they may have potential benefits in IS. Therefore, screening for ODG/R-induced PC12 cell damage related to stroke in *B. chinensis* may provide the possibility for further research on *B. chinensis* polysaccharides.

The first step in polysaccharide research is extraction, where selecting the right technique affects both the polysaccharide yield and the structural features and biological functionalities, which are essential for all subsequent experiments [18]. To gain deeper insights into the structure–activity relationship of polysaccharides, researchers commonly employ the gradient ethanol precipitation (GEP) method, which is both efficient and swift [19]. This method facilitates the preparation of highly homogeneous polysaccharide fractions. Notably, GEP significantly influences the physicochemical characteristics of polysaccharides, including molecular weight (Mw), monosaccharide composition, uronic acid content, and rheological properties, thereby ultimately impacting their biological activities [20]. Currently, Fan et al., Yao et al., Zhang et al., and others have used the GEP method to conduct related research on polysaccharides [21,22,23], enabling researchers to obtain a large number of polysaccharide fractions, screen for constituents with good activity, and conduct preliminary analysis of structure–activity relationships [21,22,23]. Nevertheless, there is a lack of studies examining the preparation, physicochemical properties, and biological activities of *B. chinensis* polysaccharides (BCPs) using the GEP approach combined with column chromatography separation technology.

In this study, four polysaccharides (BCP30, BCP50, BCP70, and BCP90) were obtained from BCPs by the gradient ethanol (30–90%) precipitation method. Subsequently, BCP30-1a, BCP50-1a, BCP70-1a, and BCP90-1a were obtained through purification and isolation. For the first time, their physicochemical properties and protective effects on ischemic stroke cell models were studied with the aim of providing a basis for the preparation and selection of highly active BCPs, the structure–activity relationship of BCPs, and broadening the exploration of active substances in BCPs.

## 2. Results and Discussion

### 2.1. Determination of the Physicochemical Characteristics

#### 2.1.1. Determination of Chemical Composition

The polysaccharide components separated from BCP30, BCP50, BCP70, and BCP90 by a DEAE-52 cellulose column and Sephadex G-200 were named BCP30-1a, BCP50-1a, BCP70-1a, and BCP90-1a in sequence (Figure 1). Their yields and chemical compositions are summarized in Table 1. Four alcohol precipitation products obtained from BCPs were successfully prepared, i.e., BCP30, BCP50, BCP70, and BCP90, with corresponding yields of 0.83 ± 0.42%, 21.47 ± 0.73%, 17.55 ± 0.81%, and 2.97 ± 0.55%. The yield of BCP50 was significantly higher (*p* < 0.05) than that of other fractions, indicating that BCP50 was the main constituent of BCPs. The final content of ethanol increased from 30% to 90% (*v*/*v*), and a small amount of protein and uronic acid was present in all four BCP components obtained by the gradient ethanol precipitation (GEP) method. Additionally, the protein content in BCP70 and BCP90 was relatively high compared to that of the other polysaccharide fractions (*p* < 0.05). This is similar to the research on Satsuma mandarin polysaccharides [21], *Sargassum fusiforme* polysaccharides [22], and *Schisandra sphenanthera* polysaccharides [23], which means that the polysaccharide constituents obtained through the GEP method will be doped with a small amount of protein and a certain amount of uronic acid.

It is worth noting that all purified and isolated BCP30-1a, BCP50-1a, BCP70-1a, and BCP90-1a contain uronic acid, which is an important active ingredient in polysaccharides [24]. Specifically, BCP50-1a exhibited a notably higher uronic acid content (27.61 ± 0.72%) compared to that of other polysaccharide fractions obtained via the GEP method, with statistical significance at *p* < 0.05. This is consistent with the findings of Fan et al., Yao et al., and Zhang et al. [21,22,23]. The uronic acid trends they studied are slightly different, which may be determined by the structure and properties of the polysaccharides. In addition, the total sugar content of BCPs after purification and isolation was higher than 85%, and no protein was detected, which laid a good material foundation for further exploration of their physicochemical structure and biological activity. These research results indicated that there were differences in the chemical composition of polysaccharides obtained through the GEP method, which suggested that controlling the ethanol concentration in polysaccharide precipitation was crucial for the chemical composition of the polysaccharide fractions and even affected their biological functions.

#### 2.1.2. Determination of FT-IR Spectroscopy

FT-IR spectroscopy has emerged as a dependable methodology for elucidating the functional groups and glycosidic bond types of polysaccharides. FT-IR was conducted to gain insight into the four types of BCP polysaccharides via their functional groups and group orientation to assess the functional group differences and similarities among them. Figure 2 shows the FT-IR spectra of BCP30-1a, BCP50-1a, BCP70-1a, and BCP90-1a. Specifically, two typical characteristic absorption bands were identified for BCPs: a broad and strong absorption band at around 3370 cm^−1^ attributed to the stretching vibration of the O-H group [25] and a small and weak band at 2930 cm^−1^ related to the stretching vibration of the C-H group [26]. The absorption peak in the vicinity of 1700 cm^−1^ was indicative of carboxyl groups, confirming the existence of uronic acid in the past investigation. Additionally, the peaks located near 1600 cm^−1^ and 1400 cm^−1^ were assigned to a water-related band and stretching vibrations of C=O or bending of C-O, respectively, during the analysis [27,28]. The spectral absorptions spanning the range of 1200 to 1000 cm^−1^ could potentially be attributed to overlapping ring vibrations, suggestive of the presence of pyranose sugars. The absorption peaks observed within the 900 to 700 cm^−1^ region may be assigned to β-glycosidic linkages and α-glycoside structures [29,30]. Overall, BCP30-1a, BCP50-1a, BCP70-1a, and BCP90-1a exhibited very similar infrared absorption peaks within the measurement wavenumber range of 4000–400 cm^−1^, indicating that their preliminary structural characteristics were similar. However, their different peak adsorption intensities indicated some differences in the structure, which may further lead to differences in their biological functions. The analysis and conclusion of this infrared spectrum are consistent with the polysaccharides of *Herba Patriniae* and *Allium sativum* L. studied by Hui et al. and Yan et al. [31,32]. These results indicate that ethanol precipitation may not alter the functional groups of BCPs.

#### 2.1.3. Determination of Molecular Weight and Polydispersity

The molecular weights (Mws) of the four BCP fractions were approximated utilizing GPC analysis. As depicted in Figure 3 and Table 2, the resulting chromatogram exhibited a single, symmetrical peak, and the polysaccharide samples displayed a polydispersity index ranging from 1.0 to 1.2, indicating a high degree of uniformity within the polysaccharide group. The molecular weights of BCP30-1a, BCP50-1a, BCP70-1a, and BCP90-1a were determined to be 198.398 kDa, 184.690 kDa, 184.556 kDa, and 184.217 kDa, respectively. Notably, an intriguing trend was observed where the retention time increased in proportion to the ethanol concentration. This trend implied that as the ethanol concentration rose, the main peak’s retention duration in each sample progressively elongated, hinting at a reduction in the sample’s molecular weight. This result is similar to the experimental results obtained by Zhang et al. [23], Hui et al. [31], and Yan et al. [32] using the GEP method to determine the Mw of polysaccharide constituents. This may be because the Mw of polysaccharides exhibits an inverse relationship with their solubility, and this solubility positively correlates with the concentration of ethanol [33,34]. It is known that the GEP method can be employed to precipitate polysaccharides with progressively lower Mws [35,36]. In addition, the Mws of BCP50-1a, BCP70-1a, and BCP90-1a differed by around 300 Da, which is not a great difference, probably due to the fact that BCPs had a unique structure leading to little difference in their solubility at high concentrations of ethanol precipitation. In any case, the biological functions of polysaccharides are closely related to their Mw, and in this study, polysaccharide fractions of BCPs with molecular weights ranging from large to small were obtained by the preparation of graded alcohol precipitation.

#### 2.1.4. Determination of the Monosaccharide Composition

The HPLC analysis revealed the composition and molar ratio of monosaccharides within BCP30-1a, BCP50-1a, BCP70-1a, and BCP90-1a, as depicted in Figure 4 and summarized in Table 2. These fractions shared an identical monosaccharide composition, namely, Man, GalA, Glc, Gal, and Ara, but the content was different. Obviously, all four polysaccharides are heteropolysaccharides, with Gal having the highest content and Man having the lowest content. These differences may be due to the varying solubility of different polysaccharides in ethanol. Thus, different ethanol concentrations did not change the monosaccharide composition of BCPs but only affected the molar ratio, consistent with the trends reported for polysaccharides from *Sagittaria sagittifolia* L [37], corn silk [38], and *Lentinus edodes* [39].

#### 2.1.5. Methylation Analysis

The determination of polysaccharide glycosidic bond types and proportions utilized GC-MS analysis of polysaccharide methylation as a crucial technique [40]. This method hinges on the retention times and mass spectrometry features of the resulting fragments. The mass spectrometry data for partially methylated sugar alcohol acetates (PMAAs) referenced the Complex Carbohydrate Structural Database (CCSD) maintained by the Complex Carbohydrate Research Centre (CCRC) at the University of Georgia, USA, alongside pertinent published information [41]. The analysis results of methylation are summarized in Figure 5 and Table 3. BCPs were composed of residues such as Man, GalA, Glc, Gal, and Ara. According to the retention time and peak area percentage of partially methylated sugar alcohol acetate, BCP30-1a, BCP50-1a, BCP70-1a, and BCP90-1a contain six identical glycosidic bonds (Man*p*-(1→, →5)-Ara*f*-(1→, →3)-Gal*p* (or GalA*p*)-(1→, →4)-Glc*p*-(1→ and →3,6)-Gal*p*-(1→) with different molar ratios. Meanwhile, compared to them, BCP30-1a had an additional glycosidic bond type of Gal*p*-(1→. From the methylation result, it could be estimated that the major chains of BCP30-1a, BCP50-1a, BCP70-1a, and BCP90-1a mainly consisted of →3)-Gal*p* (or GalA*p*)-(1→ and →4)-Glc*p*-(1→. In addition, →3)-Gal*p* (or GalA*p*)-(1→ residues were characterized as branching points of the main carbohydrate chain of four polysaccharides. The side chain terminated with Man*p*-(1→ residues, as they were the primary terminal residues of BCPs, and BCP30-1a also contained Gal*p*-(1→. It is interesting that as the ethanol concentration increased and the Mw decreased, the Man*p*-(1→ and→ →5)-Ara*f*-(1→ residues gradually decreased, which may be attributed to the GEP method endowing them with different structures.

#### 2.1.6. Scanning Electron Microscopy Analysis

SEM serves as a technique among several used to observe the advanced structure of polysaccharides, facilitating in-depth examination of their fine structures [42]. As shown in Figure 6, at a magnification of 200×, BCP30-1a, BCP50-1a, BCP70-1a, and BCP90-1a displayed more similar microstructures, characterized by fragmented flake-like structures of uneven sizes, and no discernible small pores. However, at higher magnification, BCPs exhibited distinct microstructures, primarily characterized by surface roughness accompanied by cracking of varying severity. The distinct differences in the microstructures of polysaccharides may be attributed to the ethanol concentration in polysaccharide precipitation, variations in monosaccharide compositions, glycosidic bond configurations, bond strengths, and other factors [43].

#### 2.1.7. Thermal Gravimetric Analysis

The TG and DTG curves of the BCPs are shown in Figure 7. Three stages of weight loss were observed in the process of temperature increase (Figure 7a). In the first stage, weight loss was observed between 30 °C and 227.1 °C, primarily attributed to the loss of free and bound water molecules [44]. During the second stage, weight loss occurred within the temperature range of 227.2–419.7 °C, accompanied by decarboxylation of polysaccharide carboxyl groups and the breaking of some of their linkages [45]. In the third stage, spanning from 419.8 to 800 °C, weight reduction was primarily attributed to the thermal decomposition of carbon [46]. The DTG curves indicated that the peak mass loss rate occurred at 247.1 °C for BCP30-1a, 230.6 °C for BCP50-1a, and 232.4 °C for BCP70-1a, whereas BCP90-1a showed a peak at 237.5 °C due to the highest thermal decomposition temperature (Figure 7b). Overall, BCP30-1a, BCP50-1a, BCP70-1a, and BCP90-1a exhibited good thermal stability. The polysaccharides with good thermal stability display great potential for food and pharmaceutical processing.

### 2.2. Biological Avtivity and Structure–Activity Relationship Analysis

Firstly, the CCK-8 method was used to detect the cytotoxicity of BCPs against PC12 cells at concentrations ranging from 10 µg/mL to 90 µg/mL. The results showed that BCPs exhibited relatively good cell viability at a concentration of 30 µg/mL, which led to the selection of this concentration for subsequent experimental measurements (Figure 8a). Furthermore, the viability of PC12 cells exceeded 85%, as illustrated in Figure 8b,d, suggesting that crude polysaccharides (BCP30, BCP50, BCP70, and BCP90), as well as purified and isolated polysaccharides (BCP30-1a, BCP50-1a, BCP70-1a, and BCP90-1a), exhibited no cytotoxic effects at a concentration of 30 µg/mL. The next step of the study will be to evaluate the protective effects of BCPs on OGD/R-induced ischemia–reperfusion cell models and determine their impact on cell viability. As shown in Figure 8c,e, the neuroprotective effect based on the OGD/R model of purified and isolated polysaccharides is significantly better than that of crude polysaccharides from *B. chinensis*. This means that the combination of the GEP method and column chromatography separation technology can not only prepare high-purity polysaccharide fractions but also enhance the neuroprotective effect of crude BCPs. Further, OGD/R significantly reduced the viability of PC12 cells (*p* < 0.001), but pretreatment with BCP50-1a, BCP70-1a, and BCP90-1a at 30 µg/mL significantly improved the viability of PC12 cells (*p* < 0.05), indicating that BCP50-1a, BCP70-1a, and BCP90-1a have protective effects on OGD/R-induced PC12 cells and can serve as potential active constituents for ischemic stroke (Figure 8e). However, BCP30-1a did not show any protective effect on PC12 cells, which must be due to its structural differences.

Next, the possible reasons for the different neuroprotective effects exhibited by the structural differences of BCP30-1a, BCP50-1a, BCP70-1a, and BCP90-1a will be analyzed, as the biological activity of different polysaccharides is closely associated with sugar structural configurations and conformations [47]. The structural study of the BCPs mentioned above indicates that they share similar functional groups and monosaccharide compositions but differ in chemical compositions, monosaccharide composition molar ratios, Mw, glycosidic bond types, microstructures, and thermal stability. All of these factors suggest that they may exert different cell protective effects.

The presence of charged features, such as acidic or alkaline groups, determines the neutrality and acidity of polysaccharides in their molecular structure. Among them, acidic polysaccharides have garnered significant attention for biological applications, owing to their enhanced bioactive properties compared to neutral polysaccharide fractions [48]. In this study, BCP30-1a, BCP50-1a, BCP70-1a, and BCP90-1a all contain uronic acid, indicating that they may have good biological activity. However, their differences in uronic acid content and possible connection modes also indicate differences in their activity. In addition, the structural characteristics of polysaccharides, particularly the composition and proportion of monosaccharides within the sugar ring, play a crucial role in determining their biological properties [49]. Variations in monosaccharide proportions in BCP30-1a, BCP50-1a, BCP70-1a, and BCP90-1a can alter the linearity or branching of their structure, thereby influencing solubility, polarity, stability, functionality, rheological behavior, and biological activities. The characteristics of glycosidic bonds within polysaccharides, encompassing their types, configurations, and positions, exert a notable impact on their biological activities [50]. In the current study, all four polysaccharides have Man*p*-(1→, →5)-Ara*f*-(1→, →3)-Gal*p* (or GalA*p*)-(1→, →4)-Glc*p*-(1→ and →3,6)-Gal*p*-(1→ with different molar ratios. Meanwhile, BCP30-1a contains Gal*p*-(1→, which may be one of the reasons why BCP30-1a does not have a protective effect on OGD/R-induced PC12 cells. The different microstructures of BCP30-1a, BCP50-1a, BCP70-1a, and BCP90-1a also indicate differences in their advanced structures. Finally, the Mw of polysaccharides is one of the important influential indicators of their biological activity [51]. Polysaccharides exhibiting lower Mw display reduced viscosity and enhanced solubility, whereas those with high Mw can hinder interactions with cellular receptors due to elevated viscosity or chain entanglement, resulting in diminished bioactivity [52]. Therefore, it is crucial to prepare BCPs with appropriate Mws, and the GEP method is precisely a good way to obtain polysaccharides with different Mw fragments.

## 3. Materials and Methods

### 3.1. Materials and Chemicals

The dried rhizomes of *Belamcanda chinensis* (L.) DC. originated from Yanji, located in Jilin Province, China. Verification of the samples was conducted by Professor Huizi Lv, and a voucher specimen (YB-BC-202009) has been archived at the College of Pharmacy, Yanbian University. Dextran samples of varying molecular weights, which were national drug reference materials, were obtained from both China Foods Limited and the Drug Control Institute (Beijing, China). Shanghai Aladdin Bio-Chem Technology Co., Ltd. (Beijing, China) supplied mannose (Man), ribose (Rib), rhamnose (Rha), glucosamine (GlcN), glucose (Glc), xylose (Xyl), arabinose (Ara), glucuronic acid (GlcA), galacturonic acid (GalA), galactose (Gal), and fucose (Fuc). PC12 cells (rat adrenal pheochromocytoma cells, highly differentiated) were purchased from Shangen Biotechnology Co., Ltd. (Wuhan, China). Cell culture reagents, specifically DMEM (D1152) and FBS (F7524), were sourced from Sigma Aldrich, located in St. Louis, MO, USA. The other chemical reagents used were HPLC grade or analytical grade.

### 3.2. Preparation of Polysaccharides

The rhizome of *B. chinensis* was dried, subsequently pulverized using a disintegrator, and then sieved through a 20-mesh screen to produce a fine powder. Crude polysaccharides (BCPs) were prepared by the gradient ethanol precipitation (GEP) method. Briefly, BCPs were extracted using a hot water extraction and alcohol precipitation technique; the extract parameters and conditions were as follows: temperature of 84 °C, using a liquid-to-solid ratio of 42 mL/g, and extraction for 100 min. The supernatant was collected and concentrated to 1/5 of the original volume. Anhydrous ethanol was then added to this concentrated solution until the ethanol concentration was 30%, and precipitation was carried out at 4 °C for 12 h. Centrifugation at 2000 r/min for 10 min was following by freeze-drying to obtain BCP30. Additional ethanol was introduced to the supernatant until final ethanol concentrations of 50%, 70%, and 90%, resulting in the precipitation of fractions designated as BCP50, BCP70, and BCP90, respectively. The preparation process of BCPs is shown in Figure 9, and the calculation of the polysaccharide yield is shown in Equation (1). Subsequently, the sample underwent deproteinization through the addition of 1/3 volume of Sevag reagent (chloroform: n-butyl alcohol = 4:1, *v*/*v*) [53], followed by agitation at 200 rpm for 30 min. Centrifugation at 6000 rpm for 10 min was employed to collect the supernatant, with this process being iterated until all free proteins were thoroughly eliminated. Then, dried polysaccharide fractions were redissolved in distilled water, purified by DEAE cellulose (elution conditions: 0.5 mol/L NaCl solution, 1 mL/min), Sephadex G-200 (elution conditions: distilled water, 0.3 mL/min), and dialysis (molecular weight cutoff: 8000 g/mol) to obtain BCP30-1a, BCP50-1a, BCP70-1a, and BCP90-1a.(1)$$\mathrm{polysaccharide}\;\mathrm{yield}=\frac{M_{1}}{M_{2}}$$where M_1_ (g) is the weight of dried polysaccharide powder; M_2_ (g) is the weight of raw materials

### 3.3. Determination of the Physicochemical Characteristics

#### 3.3.1. Determination of the Chemical Composition

The phenol–sulfuric acid method [54], with D-glucose as the reference, was employed to ascertain the total sugar content. The sulfuric acid–carbazole method [55], with galacturonic acid as the benchmark, was adopted to estimate the uronic acid content. Additionally, the Bradford method [56], utilizing bovine serum albumin as the standard, was used to quantify the protein content.

#### 3.3.2. Determination of FT-IR Spectroscopy

The KBr press method facilitated the determination of functional groups within 2 mg of the dried polysaccharide fractions via Fourier transform infrared spectroscopy (FTIR, Thermo Science Nicolet iS20, Thermo Fisher, Waltham, MA, USA), with a spectral range spanning from 400 to 4000 cm^−1^.

#### 3.3.3. Determination of Molecular Weight and Polydispersity

To determine the molecular weight (Mw), gel permeation chromatography (GPC, Shimadzu, Kyoto, Japan) was employed, utilizing a Shimadzu system from Japan, equipped with a refractive index detector (RID-20A) and a KS-804 column specifically designed for sugars. The specific experimental parameters were as follows: the eluent was ultrapure water, the flow rate was 1 mL/min, and the column and detector temperature was 40 °C. A calibration curve was constructed using a range of standard dextrans with varying average molecular weights. Subsequently, the molecular weight of the sample was determined by referencing this standard curve (logMw = −0.4384x + 10.89).

#### 3.3.4. Determination of Monosaccharide Composition

High-performance liquid chromatography (HPLC, Hitachi Primaide, Tokyo, Japan) offers a versatile tool for assessing the monosaccharide composition via the employment of 1-phenyl-3-methyl-5-pyrazolone (PMP) derivative methodology [57]. Briefly, the polysaccharide fractions (10 mg) were hydrolyzed with trifluoroacetic acid (120 °C, 3 h) in a hydrothermal reactor, and the excess acid in the hydrolysate was removed using anhydrous ethanol. Subsequently, PMP derivatization experiments were carried out sequentially using NaOH (0.3 mol/L, 200 µL) and PMP methanol solution (0.5 mol/L, 200 µL) in a water bath at 70 °C for 60 min, which were followed by neutralization with HCl (0.3 mol/L, 200 µL). Then, the neutral solution was extracted three times with chloroform. Finally, the aqueous layer containing derivatives labeled with PMP was analyzed by HPLC. The HPLC operational settings comprised a mobile phase consisting of a phosphate buffer solution (PBS, adjusted to pH 6.8) combined with acetonitrile at a volumetric ratio of 83:17, with a flow rate maintained at 0.8 mL/min and a detection wavelength set at 245 nm.

#### 3.3.5. Methylation Analysis

As previously described and modified in reference [58], the polysaccharide fraction (20 mg) was dissolved in dimethyl sulfoxide (12 mL) containing NaOH (240 mg) and treated with ultrasonic under nitrogen protection at room temperature until completely dissolved. Afterward, it was immersed in an ice-water bath. Subsequent to solidification, CH_3_I (3 mL) was introduced, and the mixture was subjected to sonication at ambient temperature for a duration exceeding 30 min. This step was repeated three times followed by the addition of 2 mL of distilled water to decompose the surplus CH_3_I, followed by a 48 h dialysis process. Subsequent to dialysis, the solution was subjected to a triple extraction procedure utilizing chloroform and distilled water, and then the extract was dehydrated using anhydrous Na_2_SO_4_ for a period of 24 h. The chloroform fraction was concentrated and further dried to yield the methylation products. Subsequently, the methylated product was hydrolyzed with 6 mL of 2 mol/L trifluoroacetic acid (TFA) at 110 °C for 2 h. To eliminate excess TFA, the product was co-evaporated with 5 mL of anhydrous methanol. Partially methylated monosaccharides were reduced with NaBH_4_ (10 mg/mL) at room temperature overnight. The reduced product was acetylated with 4 mL of zinc acetate hydride: pyridine (1:1, *v*/*v*) at 100 °C for 2 h. The resulting product was dissolved in chloroform and subjected to analysis utilizing a DM-5 MS column integrated within a GC-MS system (Agilent, 88605977b, Santa Clara, CA, USA). The GC-MS platform was used for analysis under the following conditions: furnace temperature initially set at 140 °C, raised to 200 °C at a rate of 2 °C/min (held for 2 min), and then raised to 280 °C at a rate of 10 °C/min (held for 5 min); helium was used as the carrier gas (1 mL/min).

#### 3.3.6. Scanning Electron Microscopy Analysis

A scanning electron microscope (Gemini SEM 360, ZEISS, Oberkochen, Germany) was employed for the detailed examination of the surface morphology and microstructural features of polysaccharides. Following the application of a thin gold coating, the lyophilized polysaccharide samples were subjected to observation under vacuum conditions, utilizing an acceleration voltage of 5 kV. Micrographs were taken at a magnifications of 200×, 500×, 10.00k×, and 15.00k×.

#### 3.3.7. Thermal Gravimetric Analysis

Thermogravimetric properties of polysaccharide fractions were determined using a thermogravimetric analyzer (TG 209 F1 Libra, NETZSCH-Gerätebau GmbH, Selb, Germany). Polysaccharides were placed in the sample dish and subjected to a temperature ramp from 30 °C to 800 °C at a rate of 20 °C/min, using the empty crucible as the basis. Derivative thermogravimetric curve profiles were obtained by differentiating the thermogravimetric values.

### 3.4. Biological Activity

PC12 cell culture: Rat pheochromocytoma cells (PC12 cells) were cultured in Dulbecco’s Modified Eagle’s Medium (DMEM) enriched with 100 units/mL penicillin, 100 mg/mL streptomycin, and 10% fetal bovine serum (FBS), maintained in 5% CO_2_ at 37 °C, with medium renewal every 48 h. When 80% to 90% of the cells were reached in the medium, the cells were digested with 0.25% trypsin, centrifuged at 1000 rpm for 5 min, and then resuspended in fresh 10% FBS DMEM medium and inoculated into cell culture dishes.

Establishment of the OGD/R model: The first step involved creating oxygen–glucose deprivation conditions by replacing the cells with sugar-free DMEM and placing them in a low oxygen or anaerobic environment for up to 4 h to simulate the pathological state of ischemia in vivo. Subsequently, the cells entered the reoxygenation stage and were placed under standard culture conditions for reoxygenation treatment to simulate the reperfusion process after ischemia. During this stage, the cells were further cultured for 24 h.

CCK-8 method detection: After 2 h of treatment, PC12 cells were subjected to OGD/R modeling. After continued culture for 24 h, 10 μL of CCK-8 reagent was added to each well and incubated in a dark incubator for 1–2 h. Then, the OD value was measured at a wavelength of 450 nm using a microplate reader (Infinite 200 PRO, Tecan, Männedorf, Switzerland). Finally, using the OD values of the control group as a reference, the cell viability rates of BCPs before and after separation and purification were calculated separately (Equation (2)).(2)$$\mathrm{Cell}\;\mathrm{viability}=\frac{A_{s}-A_{b}}{A_{c}-A_{b}}$$
where A_s_ is the absorbance of the experimental group; A_b_ is the absorbance of the blank group; and A_c_ is the absorbance of the control group.

### 3.5. Statistical Analysis

All experimental measurements were replicated three times, and their outcomes were presented in the form of mean values accompanied by standard deviations (SDs). The statistical evaluations and Pearson correlation analyses were facilitated by SPSS 17.0, ANOVA, and GraphPad Prism 6.0, and *p*-values < 0.05 were used to determine significance.

## 4. Conclusions

Currently, there is limited research on the polysaccharides derived from *Belamcanda chinensis* (L.) DC. This study employed alcohol gradient ethanol precipitation combined with column chromatography separation technology to obtain four polysaccharides (BCP30-1a, BCP50-1a, BCP70-1a, and BCP90-1a), with Gal identified as the primary monosaccharide component. Subsequently, the physicochemical properties and structural characteristics of BCPs, as well as the protective effect of these fractions on OGD/R-induced PC12 cell injury in vitro, were comprehensively analyzed. The results showed that all four polysaccharides contained β-glycosidic and α-glycosidic linkages, with a molecular weight of approximately 180 kDa. They comprised varying proportions of glycosidic linkages such as Man*p*-(1→, →5)-Ara*f*-(1→, →3)-Gal*p* (or GalA*p*)-(1→, →4)-Glc*p*-(1→ and →3,6)-Gal*p*-(1→. These polysaccharides exhibited diverse microstructures and demonstrated good thermal stability. In addition, BCP50-1a, BCP70-1a, and BCP90-1a have been found to exhibit protective effects against the injury that OGD/R induces in PC12 cells, suggesting that they could potentially be developed into active ingredients for the treatment of ischemic stroke-related diseases. Our study not only prepared BCPs of high purity and varying molecular weights using ethanol precipitation and column chromatography but also expanded the known pharmacological effects of BCPs, thereby providing a basis for further exploring their potential applications in the research field of ischemic stroke. Although this study validated the protective effect of BCP50-1a, BCP70-1a, and BCP90-1a on PC12 cells, our experimental design has certain limitations. It is worth noting that we will conduct in-depth research on BCP70-1a and BCP90-1a in the future and investigate their roles in vivo in order to gain a more comprehensive understanding of the protective mechanisms of BCPs.

## Figures and Tables

**Figure 1 molecules-30-00998-f001:**
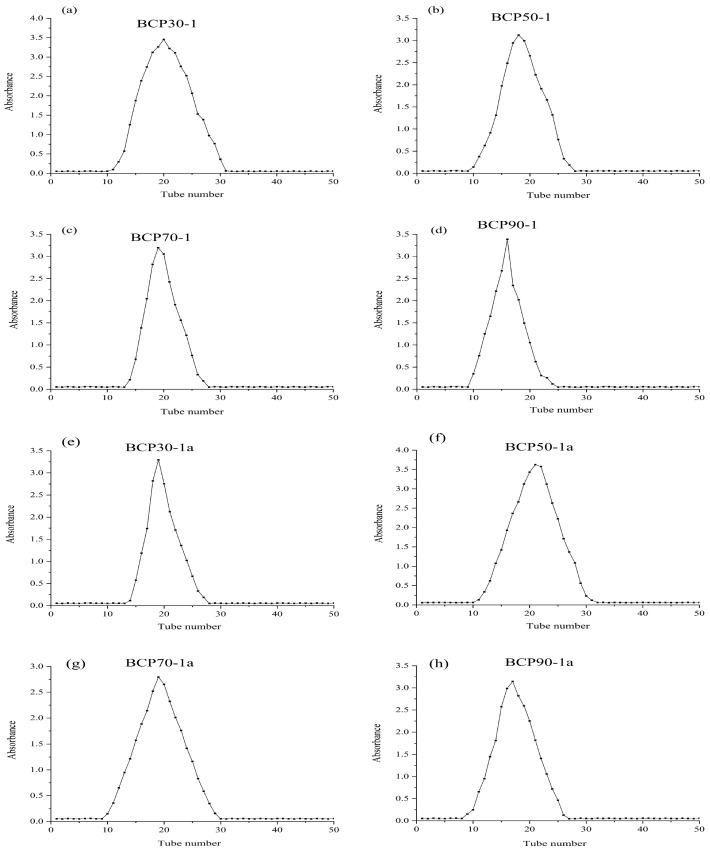
Elution curves of BCP30, BCP50, BCP70, and BCP90 on DEAE-52 cellulose (**a**–**d**); Elution curves of BCP30-1, BCP50-1, BCP70-1, and BCP90-1 on Sephadex G-200 (**e**–**h**).

**Figure 2 molecules-30-00998-f002:**
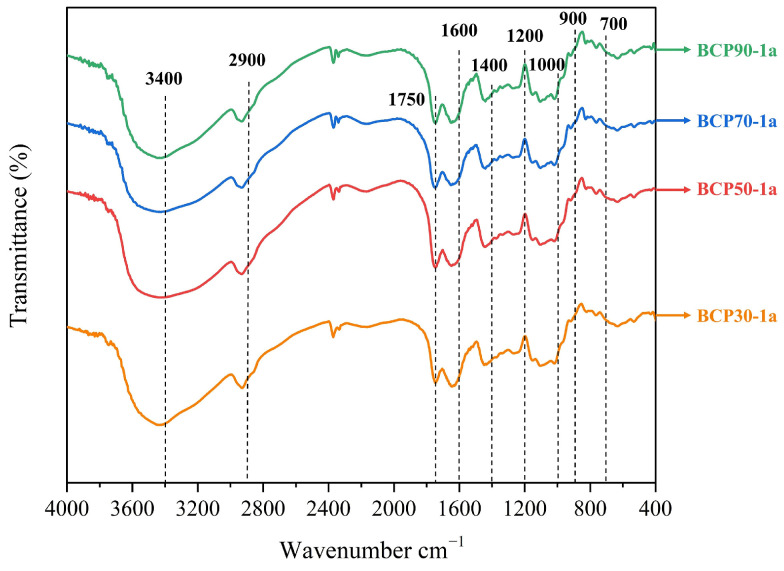
FT-IR spectra of BCP30-1a, BCP50-1a, BCP70-1a, and BCP90-1a.

**Figure 3 molecules-30-00998-f003:**
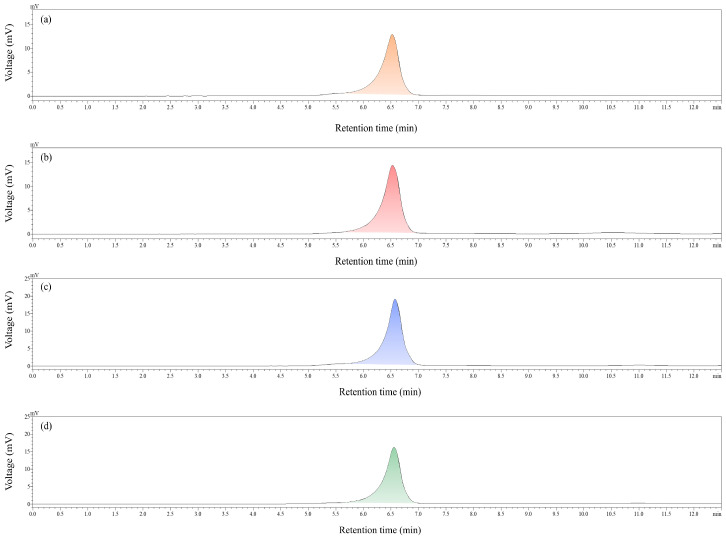
GPC traces of BCP30-1a (**a**), BCP50-1a (**b**), BCP70-1a (**c**), and BCP90-1a (**d**).

**Figure 4 molecules-30-00998-f004:**
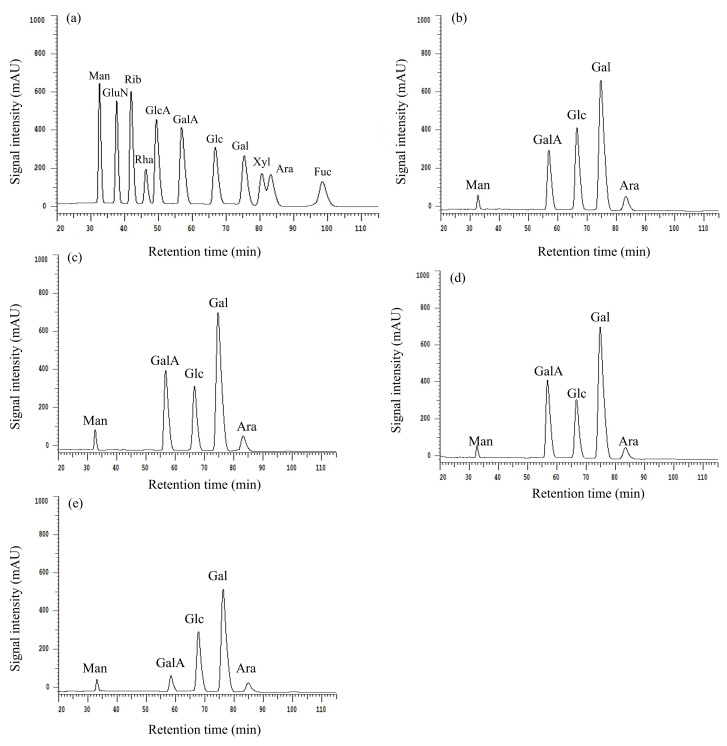
HPLC chromatograms of monosaccharide standard samples (**a**), BCP30-1a (**b**), BCP50-1a (**c**), BCP70-1a (**d**), and BCP90-1a (**e**).

**Figure 5 molecules-30-00998-f005:**
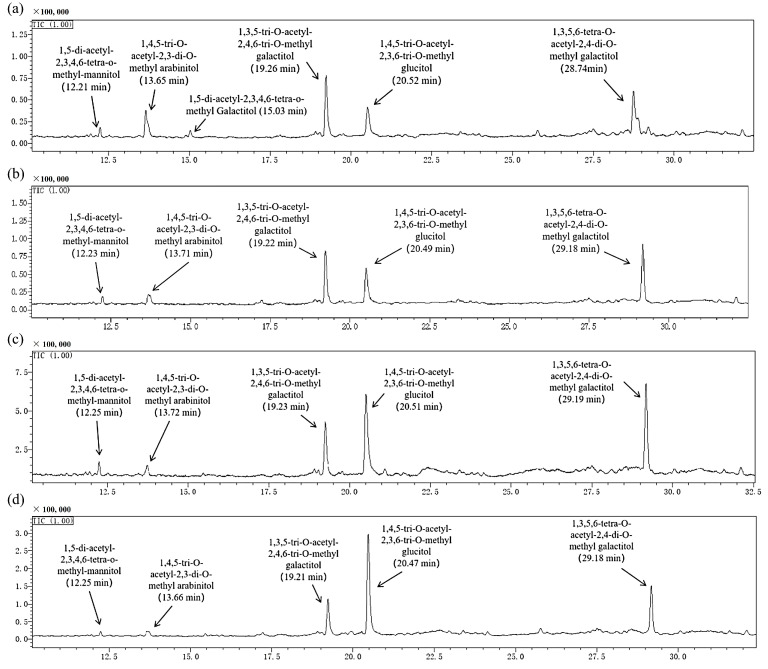
Total ion profile of partially methylated alditol acetates of BCP30-1a (**a**), BCP50-1a (**b**), BCP70-1a (**c**), and BCP90-1a (**d**).

**Figure 6 molecules-30-00998-f006:**
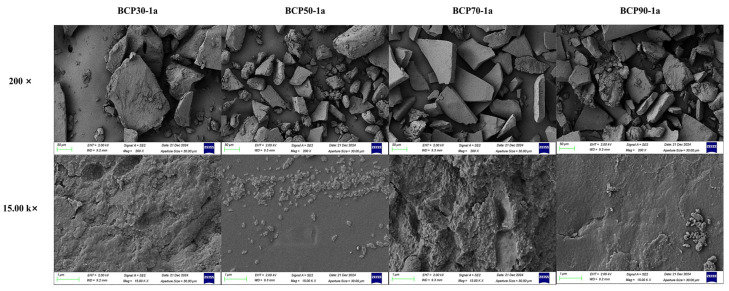
Scanning electron microscopy analysis of BCPs; the magnification was 200× and 15.00k×, respectively.

**Figure 7 molecules-30-00998-f007:**
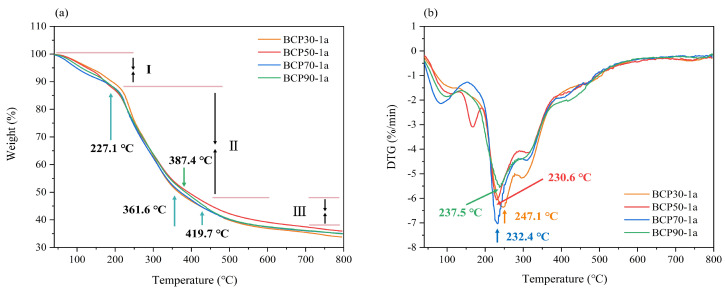
TG (**a**) and DTG (**b**) curves of BCP30-1a, BCP50-1a, BCP70-1a, and BCP90-1a (I: the first stage; II: the second stage; III: the third stage).

**Figure 8 molecules-30-00998-f008:**
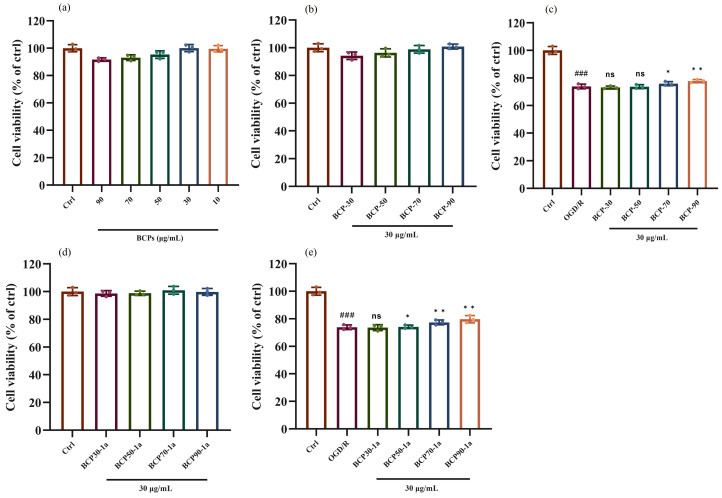
The cytotoxicity detection of BCPs on PC12 cells was tested by the CCK-8 method (**a**); The cytotoxicity detection (**b**) and cytoprotective effect (**c**) of four polysaccharides (BCP30, BCP50, BCP70, and BCP90) on PC12 cells; The cytotoxicity detection (**d**) and cytoprotective effect (**e**) of four polysaccharides (BCP30-1a, BCP50-1a, BCP70-1a, and BCP90-1a) on PC12 cells; ^ns^ *p* > 0.05, * *p* < 0.05, ** *p* < 0.01, ^###^
*p* < 0.001.

**Figure 9 molecules-30-00998-f009:**
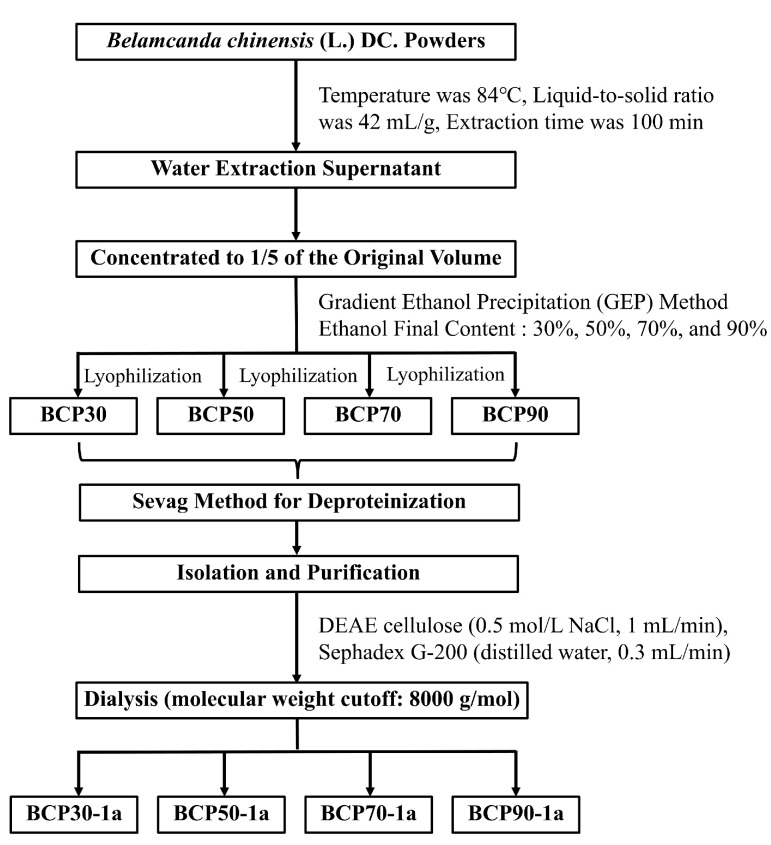
The preparation process of BCPs.

**Table 1 molecules-30-00998-t001:** Comparison of the properties of BCPs.

	BCP30	BCP50	BCP70	BCP90
Yield (%)	0.83 ± 0.42 ^d^	21.47 ± 0.73 ^a^	17.55 ± 0.81 ^b^	2.97 ± 0.55 ^c^
Total sugar content (%)	36.33 ± 0.77 ^a^	32.98 ± 0.61 ^b^	21.14 ± 0.73 ^c^	19.65 ± 0.66 ^d^
Protein content (%)	2.33 ± 0.46 ^c^	3.18 ± 0.73 ^bc^	3.96 ± 0.67 ^ab^	4.54 ± 0.62 ^a^
Uronic acid content (%)	12.25 ± 0.63 ^d^	21.71 ± 0.69 ^a^	19.87 ± 0.72 ^b^	3.65 ± 0.72 ^c^
	**BCP30-1a**	**BCP50-1a**	**BCP70-1a**	**BCP90-1a**
Total sugar content (%)	87.52 ± 0.69 ^a^	85.86 ± 0.73 ^b^	86.78 ± 0.92 ^ab^	87.46 ± 0.73 ^a^
Protein content (%)	—	—	—	—
Uronic acid content (%)	18.13 ± 0.88 ^c^	27.61 ± 0.72 ^a^	25.06 ± 0.82 ^b^	8.74 ± 0.70 ^d^

Note: Different letters indicate significant difference (*p* < 0.05); “—” indicates not detected.

**Table 2 molecules-30-00998-t002:** Molecular weight and monosaccharide composition of BCPs.

	BCP30-1a	BCP50-1a	BCP70-1a	BCP90-1a
Molecular weight (Da)	198.398	184.690	184.556	184.217
M_W_/M_n_	1.13	1.08	1.12	1.10
Monosaccharide composition (molar ratio)				
Man	1.00	1.00	1.00	1.00
GalA	7.35	8.39	12.23	2.21
Glc	15.14	7.60	11.85	13.04
Gal	20.26	15.72	24.04	18.74
Ara	1.79	1.38	1.70	1.46

**Table 3 molecules-30-00998-t003:** The methylation analysis results of BCPs.

Linkage Patterns	Retention Time (min)	Molar Percentage Ratio (%)
BCP30-1a		
Man*p*-(1→	12.21	5.24
→5)-Ara*f*-(1→	13.65	15.24
Gal*p*-(1→	15.03	3.33
→3)-Gal*p* (or GalA*p*)-(1→	19.26	30.31
→4)-Glc*p*-(1→	20.52	19.92
→3,6)-Gal*p*-(1→	28.74	25.96
BCP50-1a		
Man*p*-(1→	12.23	3.82
→5)-Ara*f*-(1→	13.71	7.53
→3)-Gal*p* (or GalA*p*)-(1→	19.22	36.82
→4)-Glc*p*-(1→	20.49	21.54
→3,6)-Gal*p*-(1→	29.18	30.29
BCP70-1a		
Man*p*-(1→	12.25	2.76
→5)-Ara*f*-(1→	13.72	3.03
→3)-Gal*p* (or GalA*p*)-(1→	19.23	20.53
→4)-Glc*p*-(1→	20.51	38.13
→3,6)-Gal*p*-(1→	29.19	35.55
BCP90-1a		
Man*p*-(1→	12.25	0.64
→5)-Ara*f*-(1→	13.66	1.06
→3)-Gal*p* (or GalA*p*)-(1→	19.21	22.45
→4)-Glc*p*-(1→	20.47	49.43
→3,6)-Gal*p*-(1→	29.18	26.42

## Data Availability

The data presented in this study are available on request from the corresponding author.

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
