# Peer review of "Physicochemical Properties and Cytoprotective Effects on PC12 Cells of Polysaccharides from *Belamcanda chinensis* (L.) DC. Obtained via a Gradient Ethanol Precipitation Method"

_molecules, 2025, doi:10.3390/molecules30050998_

Round 1
Reviewer 1 Report
Comments and Suggestions for Authors
The article focuses on the physicochemical properties and cytoprotective effects of BCPs extracted via the GEP method. However, it is not suitable for publication at current status and requires revisions.
The abstract needs to be rewritten to make it more organized and logical.
In the introduction section, the author should provide a more detailed review of the current research progress.
The author should declare the equation in the experiment part of how the extraction yield, as well as the before isolation and purification yield, were calculated. The author should re-explain the relevant results of the extraction yield in the results and discussion part. Table 1 was confusing; the author should make it more clear. For the data in table 1, there was no significant analysis of different BCPs.
In the experimental methods section, a comparison of the functions of different polysaccharide components and crude polysaccharides should be conducted to determine whether purification has reduced efficacy.
The effective information in Figure 1 is not considerable; Figure 1 should encompass the overall extraction process. The author needs to redraw Figure 1 to present more and more effective information.
Table 2, the author should explain what the numbers of the monosaccharide composition represent.
The results of the monosaccharide composition and molecular weight testing lack statistical analysis.
The figures and tables throughout the manuscript need some degree of integration and reanalysis. In the results and discussion sections, there is an excessive focus on stating the results, while analysis of the results and comparisons with existing research are relatively limited. The analysis of the reasons behind the results is superficial and needs to be strengthened.
Format mistakes, line253, 274, 287-288, 325-326, 341, the author should check the whole manuscript more carefully.
Figure 7 The scale bar in the SEM image is difficult to see clearly.
Line361, how the correlations were provided by the previous research should be stated.
Comments on the Quality of English Language
The English could be improved to more clearly express the research.
Reviewer 2 Report
Comments and Suggestions for Authors
In this study, Duan et al. investigated the physicochemical properties and cytoprotective effects of polysaccharides from Belamcanda chinensis (L.) DC. on PC12 cells. The results indicated that BCP50-1a, BCP70-1a, and BCP90-1a polysaccharides exhibited significant protective effects against ischemic stroke-induced cell injury due to their unique structural characteristics. The study was logically structured, the methodology was clear, and the results were well articulated. Here are some comments on this study:
1. Lines 47-49, it is recommended that the authors could explain in more detail the reasons why it was studied in the OGD/R model and not in other models.
2. Lines 88-89, “Belamcanda chinensis (L.) DC”, is there an expert to identify it?
3. Line 346 “30 µg/mL”, would the authors explain why 30 µg/mL was set?
4. Section 3.2, it would be slightly less convincing and credible if the biological activity was evaluated only by cell viability changes.
5. Lines 395-396, “relevant to ischemic stroke”, this conclusion seems a bit far-fetched. It is suggested that the conclusion could be toned down.
Reviewer 3 Report
Comments and Suggestions for Authors
The manuscript is well written and the data presentation is adequate. I am not sure about the industrial implementation of these polysaccharide fractions, as for use in modern medicine, but this may be another story. Some of the headings are wrongly placed, but this will be corrected upon editorial processing I assume.
Reviewer 4 Report
Comments and Suggestions for Authors
This paper by Duan et al. examined Physicochemical properties and cytoprotective effects on PC12 cells of polysaccharides from Belamcanda chinensis (L.) DC. obtained via gradient ethanol precipitation method. The structure of the paper is rigorous and the content of the experiment is complete. However, some issues still need to be resolved before publication, which are as follows:
1. The experimental methods are simple to describe in detail, including sample pretreatment, cell survival formulas, etc.
2. The introduction part is novel from the perspective of traditional Chinese medicine, but the second half of the experiment studies the recovery of ischemia-reperfusion at the cellular level, and it is suggested to add the description of the disease at the mechanism level.
3. Why is the polydispersity index of a sample determined.
4. SEM doesn't have to choose between four multiples. Two multiples are enough.
5. Please explain in detail the text in lines 353 to 354.
6. There is no difference in the thermal stability of these samples. There is something wrong with the statement on lines 383 to 384.
7. If it is necessary to continue the research on the extracted polysaccharide in the future, which sample will you choose for the follow-up experiment after you have done so?
8. The conclusions given by the author are too broad and need to be refined.
Comments on the Quality of English LanguageEnglish should be improved.
Round 2
Reviewer 1 Report
Comments and Suggestions for Authors
The auhtors have answered all the comments.